# Piperine Induces Apoptosis and Autophagy in HSC-3 Human Oral Cancer Cells by Regulating PI3K Signaling Pathway

**DOI:** 10.3390/ijms241813949

**Published:** 2023-09-11

**Authors:** Eun-Ji Han, Eun-Young Choi, Su-Ji Jeon, Sang-Woo Lee, Jun-Mo Moon, Soo-Hyun Jung, Ji-Youn Jung

**Affiliations:** 1Laboratory Animal Science, Department of Companion, Kongju National University, Yesan-gun 32439, Republic of Korea; suranhan910@gmail.com (E.-J.H.); secure801@gmail.com (E.-Y.C.); suzy4099@gmail.com (S.-J.J.); leesw9748@gmail.com (S.-W.L.); moonjunmo541@gmail.com (J.-M.M.); jsh0019y@gmail.com (S.-H.J.); 2Research Institute for Natural Products, Kongju National University, Yesan-gun 32439, Republic of Korea

**Keywords:** piperine, apoptosis, autophagy, PI3K/Akt/mTOR pathway, oral cancer, anticancer effects

## Abstract

Currently, therapies for treating oral cancer have various side effects; therefore, research on treatment methods employing natural substances is being conducted. This study aimed to investigate piperine-induced apoptosis and autophagy in HSC-3 human oral cancer cells and their effects on tumor growth in vivo. A 3-(4,5-dimethylthiazol-2-yl)-2,5-diphenyltetrazolium bromide assay demonstrated that piperine reduced the viability of HSC-3 cells and 4′,6-diamidino-2-phenylindole staining, annexin-V/propidium iodide staining, and analysis of apoptosis-related protein expression confirmed that piperine induces apoptosis in HSC-3 cells. Additionally, piperine-induced autophagy was confirmed by the observation of increased acidic vesicular organelles and autophagy marker proteins, demonstrating that autophagy in HSC-3 cells induces apoptosis. Mechanistically, piperine induced apoptosis and autophagy by inhibiting the phosphatidylinositol-3-kinase (PI3K)/protein kinase B/mammalian target of rapamycin pathway in HSC-3 cells. We also confirmed that piperine inhibits oral cancer tumor growth in vivo via antitumor effects related to apoptosis and PI3K signaling pathway inhibition. Therefore, we suggest that piperine can be considered a natural anticancer agent for human oral cancer.

## 1. Introduction

Oral cancer is the most common malignant tumor of the head and neck worldwide. Its incidence in 2020 was 377,713 cases, indicating a rapid increase in prevalence compared to the 185,976 diagnosed cases in 1990 [1,2]. Notably, more than 90% of oral cancers are oral squamous cell carcinomas (OSCC), with high metastasis and recurrence rates [3]. Additionally, most OSCC cases are diagnosed at a progressive stage of 60–80%, with a survival rate of 30–40% [4]. Various surgical therapies, radiotherapy, and chemotherapy are available to treat OSCC. However, different side effects occur, including deformation of normal cells, dysfunction, drug resistance, and toxicity [4,5]. Thus, studies on anticancer drugs using natural substances that are non-toxic and can be obtained from nature are continuously being conducted [6,7,8].

Piperine is an amide alkaloid extracted from the fruits of the black pepper (*Piper nigrum* L.) and long pepper plants (*Piper longum* L.) [9]. Piperine is known to exhibit anti-inflammatory [10], antioxidant [11], and antimicrobial [12] effects. Piperine inhibits the growth of cancer cells and suppresses tumors in various types of cancer, including breast cancer [13] and melanoma [14]. However, despite its wide array of uses, there is a general lack of studies on its effects on the signaling pathways leading to apoptosis and its anticancer effects in OSCC.

Apoptosis is a distinct mechanism of cell death that differs from necrosis in that it forms apoptotic bodies surrounded by membranes [15]. Apoptosis is characterized by cell shrinkage, chromatin condensation, nuclear division, membrane blebbing, and DNA fragmentation [16,17]. Various genes and proteins participate in apoptosis, including B-cell lymphoma 2 (Bcl-2) and B-cell lymphoma-extra large (Bcl-xL), which prevent the removal of cell growth factors to protect cells and inhibit apoptosis [18,19]. Additionally, Bcl-2 associated X (Bax) and Bcl-2 homologous antagonist killer (Bak), members of the pro-apoptotic Bcl-2 family, accumulate at the mitochondrial outer membrane and induce apoptosis by facilitating the release of cytochrome c [20]. Poly (ADP-ribose) polymerase (PARP) is involved in DNA repair and is cleaved during apoptosis [21]. In cancer types with a high percentage of apoptosis, cell growth is relatively slow despite the high rate of mitotic division of cancer cells [16]. Furthermore, increased apoptosis can inhibit tumor growth [22]. This implies that methods to increase the rate of apoptosis can be applied to cancer treatment.

Autophagy is a mechanism by which cellular components are degraded and recycled, thereby assisting in the survival of cancer cells [23,24]. However, abnormal autophagic activity may induce autophagic apoptosis by facilitating the unnecessary degradation of key proteins and organelles in cancer cells [25]. Autophagy begins with the formation of autophagosomes that can fuse with lysosomes [26]. Beclin-1 plays a central role in the early stages of autophagosome formation [27,28], and microtubule-associated protein 1A/1B-light chain 3 (LC3) found in the membrane of autophagosome serves as an indicator of the progression of autophagy [29]. Hence, autophagic activity can be determined based on the expression of these proteins, which are closely associated with the positive and negative regulation of apoptosis [30].

The phosphatidylinositol-3-kinase (PI3K)/protein kinase B (Akt)/mammalian target of rapamycin (mTOR) pathway is critical for cell proliferation, cellular metabolism, angiogenesis, and cell cycle progression [31]. It contributes to the growth of cancer cells in various human cancer types by inhibiting apoptosis [32,33]. PI3K-induced Akt activity is vital in facilitating metastasis and inhibiting apoptosis [34,35]. Moreover, these processes promote mTOR activity to engage in protein synthesis in cancer cells, promote tumor growth [36], and control cellular responses by suppressing autophagy [37]. Thus, inhibiting the PI3K/Akt/mTOR pathway is closely associated with the induction of apoptosis and autophagy.

In this study, we aimed to confirm piperine-induced apoptosis and autophagy in oral cancer HSC-3 cells and to evaluate the relationship between these processes and the PI3K/Akt/mTOR pathway.

## 2. Results

### 2.1. Piperine Reduces the Viability of HSC-3 Cells

HSC-3 cells were treated with piperine (Figure 1A) at concentrations of 0, 50, 100, 150, and 200 μM for 24 h, and changes in cell viability were confirmed using the 3-(4,5-dimethylthiazol-2-yl)-2,5 diphenyltetrazolium bromide (MTT) assay. As a result, compared with that of the control group, the mean cell viability of HSC-3 cells was 91.27% at 50 μM, 74.96% at 100 μM, 43.75% at 150 μM, and 30.69% at 200 μM. In addition, the IC_50_ value of piperine in HSC-3 cells was 143.99 μM (Figure 1B).

### 2.2. Piperine Induces Apoptosis in HSC-3 Cells

Apoptotic bodies of HSC-3 cells induced by piperine were observed through the principle that as cell membrane permeability increases during apoptosis, a greater amount of 4′,6-diamidino-2-phenylindole (DAPI) reagent stains the nucleus. DAPI staining was performed after treating HSC-3 cells with piperine at concentrations of 0, 100, and 150 μM for 24 h. The proportion of apoptotic cells significantly increased to 9.97% in the group treated with 100 μM piperine and 13.37% in the group treated with 150 μM piperine (Figure 2A,B). Additionally, HSC-3 cells were treated with piperine at concentrations of 0, 100, and 150 μM for 24 h, stained with annexin-V and propidium iodide (PI), and analyzed via fluorescence-activated cell sorting (FACS). The apoptosis rate significantly increased to 27.15% in the group treated with 100 μM piperine and 31.08% in the group treated with 150 μM piperine (Figure 2C,D). Furthermore, the expression levels of cleaved PARP, Bax, and Bcl-2 after piperine treatment were confirmed using western blot analysis. Consequently, the expression of cleaved PARP was significantly increased in the group treated with 150 μM piperine compared with that in the control group, and the expression of Bax was significantly increased in the groups treated with 100 and 150 μM piperine. However, the expression of Bcl-2 was significantly decreased in the groups treated with 100 and 150 μM piperine compared with that in the control group (Figure 2E,F).

### 2.3. Piperine Induces Autophagy in HSC-3 Cells

Acridine orange (AO) staining was performed by treating HSC-3 cells with piperine at concentrations of 0, 100, and 150 μM for 24 h. As a result, the percentage of cells with acidic vesicular organelles (AVOs) increased in a concentration-dependent manner in the groups treated with 100 and 150 μM piperine compared with those in the control group (Figure 3A). In addition, the expression level of Beclin-1 and LC3 after piperine treatment was confirmed through western blot analysis. Both Beclin-1 and LC3-II expression was significantly increased in the groups treated with 100 and 150 μM piperine compared with their expression in the control group (Figure 3B,C). After treatment with piperine (100 μM) and the autophagy inhibitors 3-methyladenine (3-MA) and hydroxychloroquine (HCQ), the viability of HSC-3 cells was evaluated using the MTT assay. The cell viability of the group treated with 3-MA and piperine was 81.07% compared with 75.67% in the group treated with piperine alone. However, the cell viability of the group treated with HCQ and piperine was 74.77%, which was not significantly different compared with the survival rate (76.97%) of the group treated with piperine alone (Figure 3D,E). Furthermore, the expression levels of Bax and Bcl-2 following 3-MA and piperine treatment were confirmed via western blot analysis. Compared with that in the group treated with piperine alone, the expression of Bax decreased significantly in the group treated with 3-MA and piperine, whereas the expression of Bcl-2 increased significantly (Figure 3F,G).

### 2.4. Piperine Inhibits the PI3K/Akt/mTOR Pathway in HSC-3 Cells

HSC-3 cells were treated with piperine at 0, 100, and 150 μM for 24 h, and the expression levels of p-PI3K, p-Akt, and p-mTOR were determined using western blot analysis. The results demonstrated a significant concentration-dependent reduction in p-PI3K, p-Akt, and p-mTOR expression in the groups treated with piperine compared to their expression in the control group (Figure 4A,B). Furthermore, HSC-3 cells were pre-treated with the PI3K/Akt inhibitor LY294002, and cell viability was examined using the MTT assay. The results showed that the viability of cells treated with piperine alone was 76.20%, and that of cells treated with LY294002 and piperine was 70.94%, revealing a significant reduction in cell viability (Figure 4C). Additionally, the western blot assay results of cells treated with LY294002 and piperine showed a significant increase in Bax expression and a decrease in Bcl-2 expression compared with that in cells treated solely with piperine. The expression levels of p-mTOR, Beclin-1, and LC3 in cells treated with LY294002 and piperine were determined to identify the relationship between the PI3K pathway and autophagy. The results showed a significant decrease in p-mTOR expression and a significant increase in Beclin-1 and LC3-II expression in cells treated with LY294002 and piperine compared with their expression in cells treated solely with piperine (Figure 4D,E).

### 2.5. Piperine Inhibits Tumor Growth In Vivo

After tumor formation, distilled water (0 mg/kg, n = 5) or piperine (50 mg/kg, n = 5) was administered daily for 4 weeks. As a result, the growth of tumor volume and weight tended to decrease in the group administered with piperine compared to the control group (Figure 5A,B), and there was no significant difference in the body weight of mice (Figure 5C). Furthermore, hematoxylin and eosin (H&E) staining was performed to confirm liver and kidney toxicity caused by piperine administration. It was confirmed that there was no significant histopathological difference in the liver and kidney between the two groups (Figure 5D).

### 2.6. Piperine Induces Apoptosis In Vivo

The expression levels of apoptosis-related proteins and PI3K/Akt/mTOR pathway proteins in the tumor tissues of the control group and the piperine-administered group were confirmed by western blot assay. As a result, the expression of Bax increased in the group administered with piperine compared to the control group, whereas the expression of Bcl-2 decreased (Figure 6A,B). Furthermore, the expression of p-PI3K, p-Akt, and p-mTOR proteins were significantly decreased in the piperine-administered group compared to the control group (Figure 6C,D).

## 3. Discussion

Piperine is an amide alkaloid found in black pepper (*Piper nigrum* L.) and long pepper (*Piper longum* L.) [9]. Its anti-inflammatory [10], antidiabetic [38], and anti-allergic [39] effects have been verified, in addition to its antiproliferative and cytotoxic effects on various cancer cells [40,41]. However, only a few studies have investigated the anticancer effects of piperine on OSCC through the induction of apoptosis and autophagy. Thus, this study verified the anticancer effects of piperine in HSC-3 cells and identified the correlation between the PI3K/Akt/mTOR signaling pathway, apoptosis, and autophagy.

In this study, HSC-3 cells were treated with piperine, and the variation in cell viability was evaluated using the MTT assay. The cell viability decreased significantly in a concentration-dependent manner from 100 μM. In a previous study, the cervical cancer cell line KB was treated with 0, 25, 50, 100, 200, and 300 μM piperine for 24 h, and the cell viability was 90.14% at 25 μM, 76.59% at 50 μM, 52.39% at 100 μM, 25.26% at 200 μM, and 18.96% at 300 μM, compared with that in the control [42]. In another study, the breast cancer cell lines MDA-MB-231 and MCF-7 were treated with 0, 25, 50, 75, 100, 150, and 200 μM piperine for 48 h. Both cell types showed a concentration-dependent reduction in cell viability, with IC_50_ values of 173.4 µM in MDA-MB-231 cells and 111.0 µM in MCF-7 cells [43]. Thus, the piperine-induced reduction in HSC-3 cell viability observed in this study is consistent with previous studies.

Apoptosis is characterized by distinct morphological changes such as pyknosis in the nucleus, DNA fragmentation, chromatin condensation, and phosphatidylserine in the plasma membrane [44,45]. Furthermore, during the process of apoptosis, the permeability of the cell membrane increases, and apoptotic bodies become more intensely stained blue due to increased penetration of the DAPI reagent [46]. DAPI staining was performed to detect apoptotic bodies and verify whether apoptosis was caused by the piperine-induced reduction in HSC-3 cell viability. Compared with the control, the piperine-treated groups showed an increase in the number of cells with apoptotic bodies in a concentration-dependent manner. Annexin-V/PI staining was performed to quantify the level of apoptosis induced by piperine, and the piperine-treated groups exhibited a concentration-dependent increase in the rate of apoptosis compared with the control. In a previous study, HeLa human cervical cancer cells were treated with 0, 25, 50, and 100 µM piperine, and DAPI staining revealed that the percentage of apoptotic cells increased significantly to 21.33% in the group treated with 50 µM piperine and 34.66% in the group treated with 100 µM piperine, compared with that in the control [47]. Another study reported that when the human melanoma cell line SK-MEL-28 was treated with piperine at concentrations of 0, 100, 150, and 200 µM for 24 h, and annexin-V/PI staining was performed, the percentage of apoptotic cells increased to 30% in the group treated with 150 µM piperine and 45% in the group treated with 200 µM piperine, demonstrating the apoptosis-inducing effect of piperine [48]. These findings indicate that the reduced viability of HSC-3 cells by piperine was associated with the induction of apoptosis.

Bcl-2 proteins play a key role in controlling cellular decision-making during the progression of apoptosis through the mitochondria [49]. Bcl-2 was first discovered in acute lymphoblastic leukemia and has been shown to play a cytoprotective role against apoptosis, making it an apoptosis-inhibitory protein [50,51]. Conversely, Bax proteins act antagonistically to Bcl-2 proteins, increasing outer mitochondrial membrane permeability and facilitating the release of proteins from the intermembrane space to the cytosol, thereby inducing apoptosis [52]. If PARP proteins are cleaved simultaneously, the induction of apoptosis is further promoted because the damaged DNA is not properly repaired [21]. Therefore, the progression of apoptosis can be determined based on the expression of apoptosis-related proteins such as PARP, Bax, and Bcl-2. In this study, HSC-3 cells were treated with piperine at concentrations of 0, 100, and 150 µM, and a western blot assay was performed. The results showed that compared with the control group, the groups treated with piperine had a higher expression of cleaved PARP and Bax, while the expression of Bcl-2 decreased to an even lower level. In a prior study [53], the human lung cancer cell line A549 was treated with piperine at concentrations of 0, 50, 100, and 200 µg/mL. The Bax/Bcl-2 ratio increased in the piperine-treated groups compared with that in the control. Additionally, in another study [13], the human breast cancer cell line SK-BR-3 was treated with piperine at concentrations of 0, 10, 25, and 50 µM, and the induction of apoptosis was confirmed based on the increased expression of cleaved PARP. The current study showed piperine-regulated protein expression in HSC-3 cells, which aligns with previous studies. Thus, piperine induces apoptosis by regulating the expression of apoptosis-related proteins in HSC-3 cells.

Autophagy generally assists in alleviating stress and controlling homeostasis in cancer cells [23,24]. However, there are cases where autophagy promotes apoptosis in cancer cells, implying that autophagy is a double-edged sword for cancer cells [54,55]. In this study, HSC-3 cells were treated with piperine, and a concentration-dependent increase in AVOs was observed via AO staining. The increased expression of Beclin-1 and LC3-II, which are indicators of autophagy, was also confirmed. In a previous study, LNCaP and PC-3 prostate cancer cells were treated with 160 μM piperine for 24 h, and the piperine-treated groups showed increased LC3B-II expression compared with the control, demonstrating the role of piperine in autophagy induction [56]. Therefore, piperine regulates the expression of autophagy-related proteins in HSC-3 cells to induce autophagy.

Furthermore, changes in cell viability were observed in HSC-3 cells when treated with piperine and 3-MA, an inhibitor of the early stage of autophagy that prevents the formation of autophagosomes, or HCQ, which inhibits the late stage of autophagy by preventing the binding of lysosomes [57]. Compared with the treatment with piperine alone, the combination of piperine and 3-MA led to a significant increase in cell viability. Additionally, when HSC-3 cells were treated with piperine and 3-MA, the expression of Bax decreased, while the expression of Bcl-2 increased, in contrast to treatment with piperine alone. In a previous study, the human melanoma cell line A375 was treated with 3-MA and a substance with an anticancer effect, resulting in increased cell viability and colony formation compared with the treatment with the anticancer substance alone [58]. Compared with findings of previous studies, our findings revealed that the early stages of piperine-induced autophagy in HSC-3 cells could promote cellular apoptosis.

The PI3K/Akt/mTOR pathway plays a crucial role in cell survival and growth, and activity of this pathway inhibits apoptosis [32,33]. In the present study, we aimed to determine the correlation between the PI3K/Akt/mTOR pathway and apoptosis. A significant decrease in the expression levels of p-PI3K, p-Akt, and p-mTOR was observed in HSC-3 cells treated with piperine compared with those in the control. In addition, compared with the group treated with piperine alone, the group treated with piperine and LY294002, an inhibitor of PI3K/Akt, showed decreased cell viability, increased Bax expression, and decreased Bcl-2 expression. In a previous study, the breast cancer cell line MDA-MB-468 was treated with 150 μM piperine for 48 h, and the piperine-treated group showed a decrease in the expression of p-Akt compared with the control [41]. In another study where the human gastric cancer cell line SNU-16 was treated with piperine at concentrations of 0, 50, 100, and 150 μM for 18 h, the expression of p-PI3K and p-Akt was shown to decrease in a concentration-dependent manner, demonstrating an apoptosis-inducing effect by inhibiting the PI3K pathway [59]. These findings collectively indicate that the piperine-induced apoptosis of HSC-3 cells is associated with the inhibition of the PI3K/Akt/mTOR pathway. Additionally, HSC-3 cells were treated with piperine and LY294002 to determine the correlation between the PI3K pathway and autophagy, and the expression levels of the autophagy-related proteins p-mTOR, Beclin-1, and LC3 were measured. The results showed that compared with the group treated with piperine only, the group treated with piperine and LY294002 exhibited a significant decrease in p-mTOR expression but a significant increase in Beclin-1 and LC3-II expression. Activation of the mTOR protein promotes cellular metabolism to supply the necessary components for cell growth and blocks catabolic processes such as autophagy [60,61]. Hence, the decreased expression of the p-mTOR protein and the increased expression of autophagy markers indicate that autophagy is further promoted by LY294002, which inhibits the PI3K pathway.

A xenograft mouse model was also established to verify the effect of piperine in suppressing tumor growth in vivo based on its in vitro anticancer effect in HSC-3 cells. The results demonstrated that compared with the mice in the control group, mice administered with 50 mg/kg piperine exhibited a decrease in tumor volume and weight without toxic effects. In a previous study using a xenograft model of human prostate cancer, daily oral administration of 50 mg/kg piperine for 14 days suppressed tumor growth [62]. Similarly, in another study on human gastric cancer, daily oral administration of 30 mg/kg and 60 mg/kg piperine for 30 days significantly decreased tumor growth [59]. These findings indicate that piperine exerts an antitumor effect in human oral cancer cells at concentrations similar to those used in previous studies. Additionally, piperine is decomposed by acid or alkaline hydrolysis into volatile basic piperine. Furthermore, when administered orally, piperine can be absorbed rapidly and does not undergo any metabolic changes during absorption [63]. Therefore, the metabolic properties of piperine suggest that oral administration of piperine may have a significant impact on tumor growth in vivo. Moreover, tumor cells treated with piperine showed increased Bax and decreased Bcl-2 expression, whereas the expression of p-PI3K, p-Akt, and p-mTOR uniformly decreased. In a previous study using a xenograft model of human melanoma, a TUNEL assay was performed on tumor tissues obtained after treatment with 100 mg/kg piperine, confirming an increased level of apoptosis. Tumor tissues from piperine-treated mice also exhibited increased expression of cleaved caspase-3 and p-ERK1/2 in an immunohistochemistry assay, confirming their association with the MAPK pathway [14]. Therefore, it is suggested that the antitumor effect of piperine shown in this study is related to apoptosis and the PI3K pathway. However, studies focusing on the correlation of piperine with apoptosis and autophagy in vivo are ongoing, and further detailed studies on the role of piperine-related autophagy in vivo need to be performed. Furthermore, additional studies should be conducted to confirm the effect of piperine using primary tumor cells from the human oral cavity, which are more physiologically relevant than immortalized cell lines.

## 4. Materials and Methods

### 4.1. Materials and Reagents

Piperine ≥ 97%, dimethyl sulfoxide (DMSO), MTT, and DAPI were purchased from Sigma-Aldrich (St. Louis, MO, USA). Dulbecco’s modified Eagle’s medium (DMEM) and fetal bovine serum (FBS) were purchased from Welgene (Gyeongsan, Republic of Korea). Streptomycin and penicillin were purchased from Gibco BRL (Grand Island, NY, USA). Cell lysis buffer was purchased from Invitrogen (Carlsbad, CA, USA), and a fluorescein isothiocyanate (FITC)/annexin-V detection kit was purchased from Pharmingen (San Diego, CA, USA). The autophagy inhibitors 3-MA and HCQ were purchased from Selleck Chemicals LLC (Houston, TX, USA). Antibodies against PARP (rabbit, #9542), Bax (rabbit, #2772), p-PI3K (rabbit, #4228), p-Akt (rabbit, #4060), p-mTOR (rabbit, #2971), Beclin-1 (rabbit, #3738), LC3 (rabbit, #4108), rabbit IgG secondary antibodies (rabbit, #7074), and the PI3K/Akt inhibitor LY294002 were purchased from Cell Signaling Technology (Beverly, MA, USA). β-actin (mouse, sc-47778) and mouse IgG secondary antibodies (mouse, sc-516102) were purchased from Santa Cruz Biotechnology Inc. (Dallas, TX, USA), and Bcl-2 (rabbit, NB100-56098) was purchased from Novus (Littleton, CO, USA).

### 4.2. Cell Culture

The HSC-3 human oral cancer cell line was provided by Professor Sung-Dae Cho (Seoul National University, Seoul, Republic of Korea). HSC-3 cells were cultured using DMEM containing 5% FBS and 1% streptomycin/penicillin and maintained in an incubator at 37 °C and 5% CO_2_. The spent medium was replaced every 2–3 days, and the cells were subcultured when they reached 70–80% confluence.

### 4.3. MTT Assay

HSC-3 cells were seeded in a 96-well plate at a density of 1.5 × 10^4^ cells/mL, cultured for 24 h, treated with piperine (0, 50, 100, 150, and 200 μM), and incubated at 37 °C for 24 h. Prior to piperine (100 μM) treatment, the cells were pre-treated with 3-MA (2 mM), HCQ (25 μM), and LY294002 (10 μM) for 2 h at 37 °C. After removing the treatment medium and adding MTT solution (40 μL/well) for 90 min at 37 °C, DMSO (100 μL/well) was added, and the absorbance was measured at 595 nm using an ELISA reader (Bio-Rad Laboratories Inc., Hercules, CA, USA).

### 4.4. DAPI Staining

HSC-3 cells were seeded in a 60-mm dish at a density of 1 × 10^5^ cells/mL and cultured for 24 h. Subsequently, the cells were treated with piperine (0, 100, and 150 μM) and incubated at 37 °C for 24 h. The treatment medium was then removed, and the cells were fixed with 4% paraformaldehyde for 15 min. Afterward, they were stained with a DAPI solution (2 mL/dish) for 1 min at 20 °C and observed at 100× magnification using a fluorescence microscope (Zeiss Fluorescence Microscope, Thornwood, NY, USA).

### 4.5. Annexin-V/PI Staining

HSC-3 cells were seeded in a 75 cm^2^ flask and cultured for 24 h; then, the cells were treated with piperine (0, 100, and 150 μM) at 37 °C for 24 h. After removing the treatment medium, the cells were suspended using a cell scraper and centrifuged at 260× *g* for 5 min at 4 °C. Subsequently, a binding buffer (1 × 10^6^ cells/mL) was added to the cell pellet. Then, the cells were stained with annexin-V and PI for 20 min at 20 °C and analyzed using a FACSCalibur™ flow cytometer (BD Biosciences, Franklin Lakes, NJ, USA).

### 4.6. Western Blot Analysis

After seeding HSC-3 cells in a 75 cm^2^ flask and culturing them for 24 h, piperine (0, 100, and 150 μM) was added, and the cells were incubated at 37 °C for 24 h. Then, the cells were detached using trypsin-EDTA and centrifuged at 260× *g* for 5 min at 4 °C. Next, the cell pellet was treated with a cell lysis buffer (1 × 10^6^ cells/80 μL) for 20 min at 4 °C, and the supernatant obtained via centrifugation at 15,920× *g* for 5 min at 4 °C was used as the cell lysate. The collected proteins were quantified using a Bradford protein assay (Bio-Rad Laboratories Inc., Hercules, CA, USA), separated using 12% sodium dodecyl sulfate-polyacrylamide gel electrophoresis (SDS-PAGE), and transferred to a nitrocellulose membrane. The membrane was blocked with 5% skim milk or 5% bovine serum albumin (BSA) at 20 °C for 2 h. Then, anti-PARP (1:1000), anti-Bax (1:700), anti-Bcl-2 (1:1000), anti-p-PI3K (1:500), anti-p-Akt (1:500), anti-p-mTOR (1:1000), anti-Beclin-1 (1:2000), anti-LC3 (1:700), and anti-β-actin (1:2000) antibodies were added, and the membrane was incubated overnight at 4 °C. After washing, the membrane was treated with anti-mouse IgG and anti-rabbit IgG secondary antibodies for 2 h at 20 °C, and the protein expression levels were determined using enhanced chemiluminescence (ECL) detection reagents (Pierce, Rockford, IL, USA). Protein expression levels were quantified using ImageJ Launcher software version 1.52a (NCBI).

### 4.7. AO Staining

HSC-3 cells were seeded on a 60 mm dish at 1 × 10^5^ cells/mL, cultured for 24 h, and treated with piperine (0, 100, and 150 μM) at 37 °C for 24 h. The treated medium was removed, and cells were fixed by adding 4% paraformaldehyde for 15 min at 20 °C. After removing paraformaldehyde and staining with AO solution (5 μg/mL) at 20 °C for 10 min, it was observed under a fluorescence microscope at 100× magnification.

### 4.8. Establishment of Xenograft

BALB/c nude female mice (18–22 g, 4 weeks age) were purchased from Nara Biotec (Seoul, Republic of Korea). Animal experiments were conducted according to the guidelines of Kongju National University’s Institutional Animal Care and Use Committee (Chungcheongnam-do, Republic of Korea, Approval number: KNU_2023-05). In addition, animal experiments were conducted at the SPF Animal Laboratory at Kongju National University. Mice were maintained under a 12 h light/dark cycle at the specified temperature (23 ± 3 °C) and humidity (40 ± 10%). To establish the xenograft model, 1 × 10^7^ cells/mL of HSC-3 cells were added to PBS containing 20% FBS and injected subcutaneously into both shoulders of mice to form tumors. After that, the solid tumor was excised, and cube-shaped tumor sections with a cut surface length of 3 mm were implanted subcutaneously into the shoulders of the experimental mice. After 20 days, the mice with tumor growth were randomly divided into the control group (distilled water, n = 5) and the piperine group (50 mg/kg, n = 5). Distilled water or piperine (50 mg/kg) was orally administered daily for 4 weeks [62,64], and the tumor volume was measured daily using Vernier calipers (Mitutoyo, Kawasaki, Japan) and calculated using the following formula: Volume (mm^3^) = 0.5 × (wide)^2^ × length. At the end of the experiment, the mice were sacrificed using CO_2_ gas (30% per min, 3 min), and the tumor weight was measured.

### 4.9. H&E Staining

The liver and kidneys of mice were excised, fixed with 10% formaldehyde, and then embedded in paraffin. Thereafter, the paraffin block was cut to 4 μm sections and subjected to H&E staining, which was observed at 400× magnification using an optical microscope (BX41; Olympus Co., Tokyo, Japan).

### 4.10. Statistical Analysis

MTT analysis was performed using Bio-Rad’s MPM6 software version 6.1, and western blot bands were quantified using ImageJ software version 1.52a. The results are presented as the mean and standard deviation (SD). After conducting one-way analysis of variance (ANOVA) for comparisons between groups, Dunnett’s *t*-test was used, and statistical significance was indicated at *p* < 0.05 compared to the control group or the piperine-treated group.

## 5. Conclusions

This study showed that piperine inhibits the PI3K/Akt/mTOR signaling pathway in oral cancer HSC-3 cells while also inducing apoptosis and reducing cell viability in a concentration-dependent manner. Furthermore, piperine-induced autophagy leads to apoptosis, and the level of piperine-induced autophagy can be further increased by inhibiting the PI3K pathway. Additionally, the in vivo inhibitory effect of piperine on tumor growth was verified, which, consistent with its in vitro effect, was shown to be associated with the induction of apoptosis via the inhibition of the PI3K pathway. Therefore, the results of this study suggest that piperine has the potential to serve as a natural anticancer agent against human oral cancer.

## Figures and Tables

**Figure 1 ijms-24-13949-f001:**
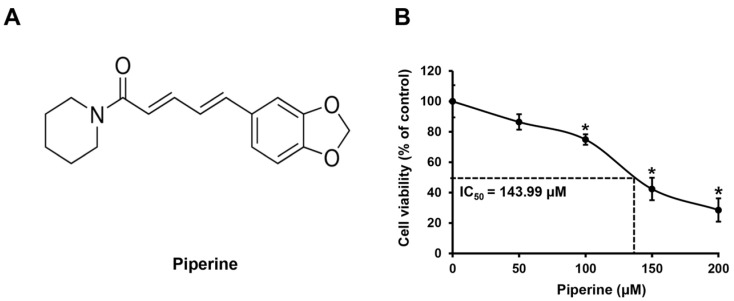
Effects of piperine on inhibition of HSC-3 cell viability. (**A**) The chemical structure of piperine. (**B**) Cell viability of HSC-3 cells treated with piperine (0, 50, 100, 150, and 200 µM for 24 h). Data are presented as mean and standard deviation (SD) for three samples. * *p* < 0.05, compared with the control group.

**Figure 2 ijms-24-13949-f002:**
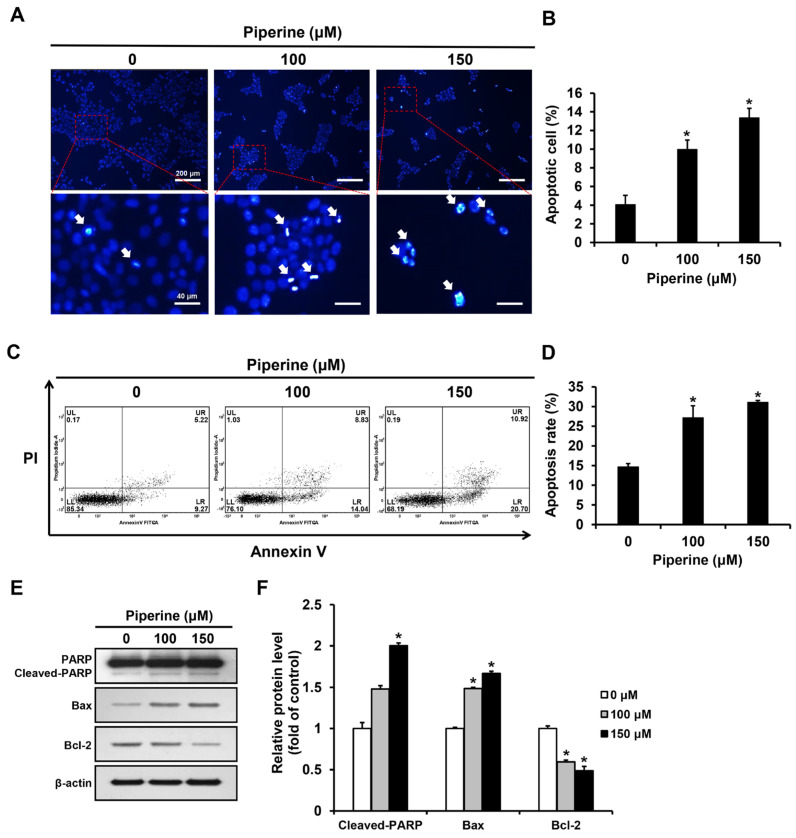
Effects of piperine on induction of apoptosis in HSC-3 cells. (**A**,**B**) Fluorescence microscopic images of HSC-3 cells treated with piperine (100 and 150 µM for 24 h) and stained with DAPI. The arrows indicate DAPI-positive HSC-3 cell. (**C**,**D**) FACS results of HSC-3 cells treated with piperine (100 and 150 µM for 24 h) and were double stained with annexin-V/PI. (**E**,**F**) Expression levels of PARP, Bax and Bcl-2 in HSC-3 cells after piperine (100 and 150 µM for 24 h) treatment. β-actin was used as loading control, and the quantification was performed using ImageJ. Data are presented as mean and SD for three samples. * *p* < 0.05, compared with the control group.

**Figure 3 ijms-24-13949-f003:**
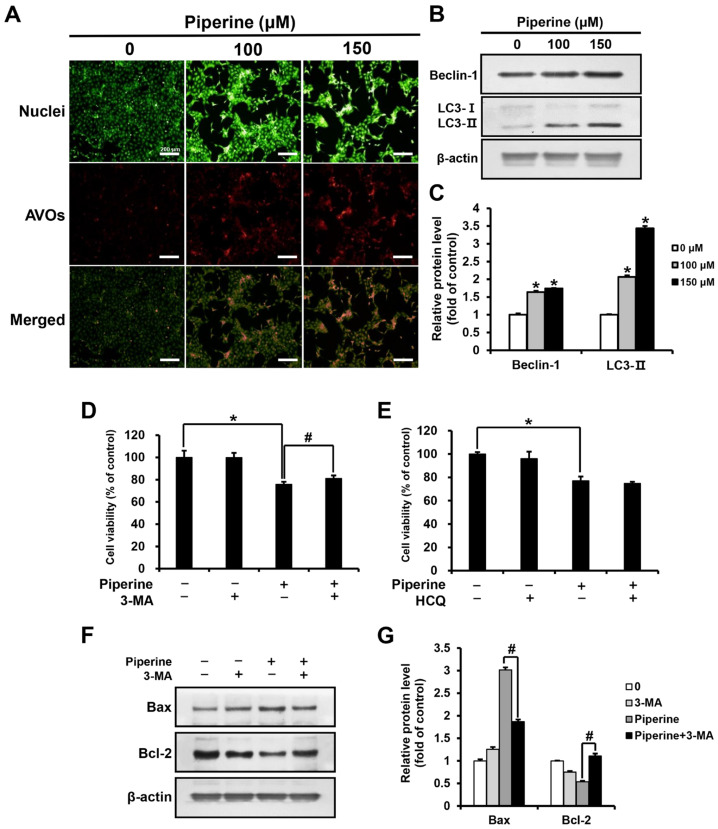
Effects of piperine on the induction of autophagy in HSC-3 cells. (**A**) Fluorescence microscopic images of HSC-3 cells treated with piperine (100 and 150 µM for 24 h) and stained with AO. (**B**,**C**) Expression levels of Beclin-1 and LC3 in HSC-3 cells after piperine (100 and 150 µM for 24 h) treatment. Cell viability of HSC-3 cells pretreated with (**D**) 3-MA (2 mM) or (**E**) HCQ (25 μM) for 2 h, followed by treatment with piperine (100 µM for 24 h). (**F**,**G**) Expression levels of Bax and Bcl-2 proteins in HSC-3 cells pretreated with 3-MA (2 mM for 2 h), followed by treatment with piperine (100 µM for 24 h). β-actin was used as loading control, and the quantification was performed using ImageJ. Data are presented as mean and SD for three samples. * *p* < 0.05, compared with the control group; ^#^
*p* < 0.05, compared with the piperine treatment group.

**Figure 4 ijms-24-13949-f004:**
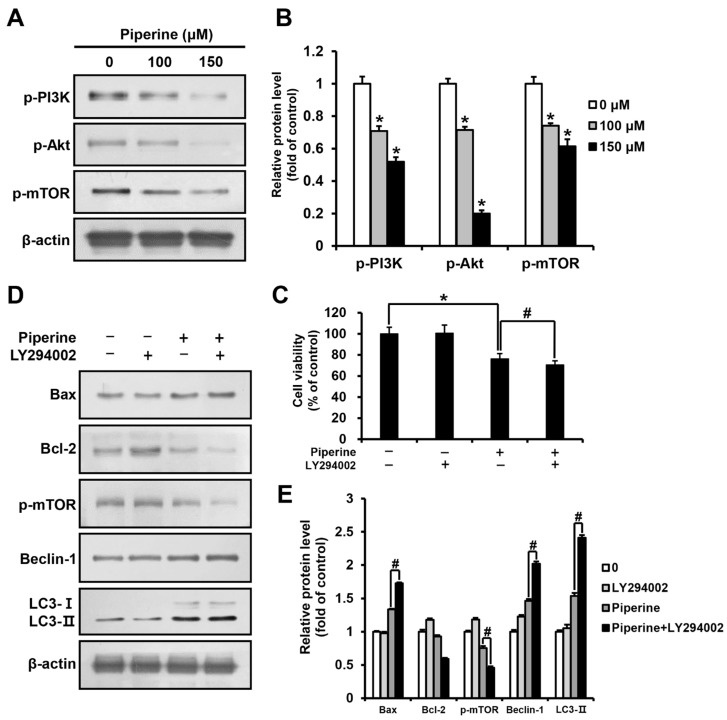
Effects of piperine on the induction of apoptosis and autophagy through the PI3K/Akt/mTOR pathway in HSC-3 cells. (**A**,**B**) Expression levels of p-PI3K, p-Akt and p-mTOR proteins in HSC-3 cells after piperine (100 and 150 µM for 24 h) treatment. (**C**) Cell viability of HSC-3 cells pretreated with LY294002 (10 μM for 2 h) followed by treatment with piperine (100 μM for 24 h). (**D**,**E**) Expression levels of Bax, Bcl-2, p-mTOR, Beclin-1 and LC3 proteins in HSC-3 cells pretreated with LY294002 (10 µM for 2 h), followed by treatment with piperine (100 µM for 24 h). β-actin was used as loading control, and the quantification was performed using ImageJ. Data are presented as mean and SD for three samples. * *p* < 0.05, compared with the control group; ^#^
*p* < 0.05, compared with the piperine treatment group.

**Figure 5 ijms-24-13949-f005:**
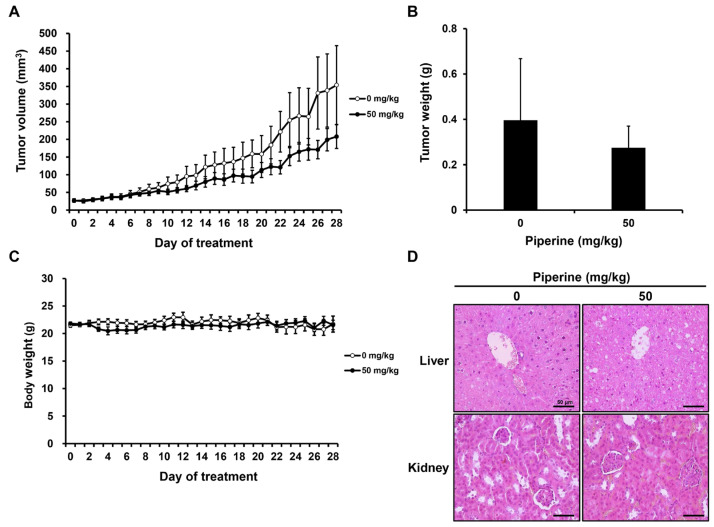
Effects of piperine on tumor growth inhibition in vivo. Nude mice bearing HSC-3 cells as xenograft models were treated with piperine (0, 50 mg/kg daily for 4 weeks), and (**A**) tumor volume, (**B**) tumor weight and (**C**) body weight were measured. (**D**) H&E staining performed for the liver and kidney.

**Figure 6 ijms-24-13949-f006:**
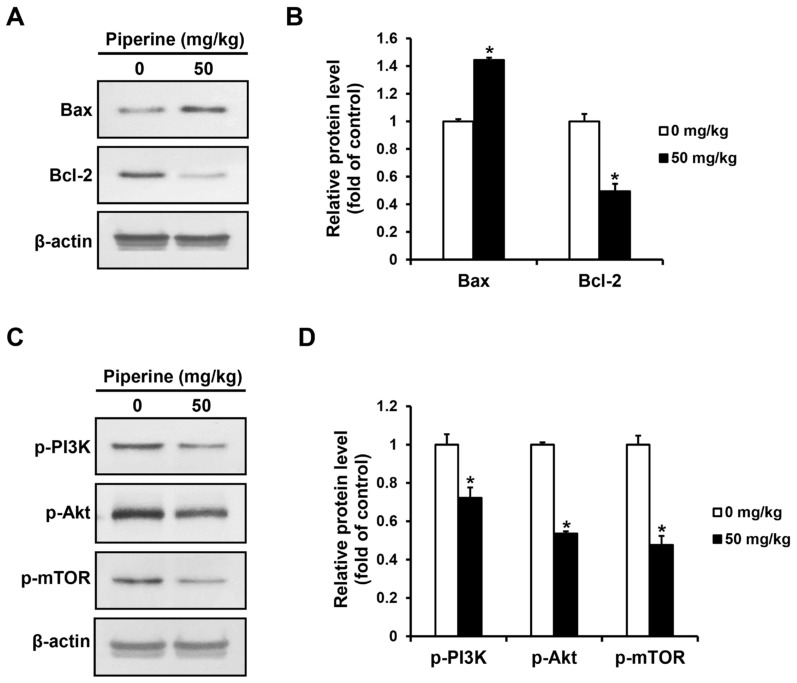
Effects of piperine-induced apoptosis through the PI3K signaling pathway in oral cancer tumor. Expression levels of (**A**,**B**) Bax, Bcl-2, (**C**,**D**) p-PI3K, p-Akt, and p-mTOR proteins in oral cancer tumors after piperine (0, 50 mg/kg daily for 4 weeks) treatment. β-actin was used as loading control, and the quantification was performed using ImageJ. Data are presented as mean and SD for three samples. * *p* < 0.05, compared with the control group.

## Data Availability

The data that support the findings of this study are available from the corresponding author upon reasonable request.

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
