# Peer review of "Piperine Induces Apoptosis and Autophagy in HSC-3 Human Oral Cancer Cells by Regulating PI3K Signaling Pathway"

_ijms, 2023, doi:10.3390/ijms241813949_

Round 1

Reviewer 1 Report

Piperine appears to exert its antiproliferative and cytotoxic effects on various cancer cells by inducing apoptosis. In this research paper, the authors examined the potential of piperine, a natural compound, as an anticancer agent for treating human oral cancer, and explored into its underlying molecular mechanisms.

This study demonstrated that piperine effectively hinders the PI3K/Akt/mTOR signaling pathway in HSC-3 oral cancer cells, leading to apoptosis and a dose-dependent decrease in cell viability. Furthermore, piperine-induced autophagy contributes to apoptosis, and this autophagic response can be enhanced by PI3K pathway inhibition. Additionally, the study confirmed the in vivo tumor growth inhibition by piperine, which, in line with the in vitro findings, is attributed to apoptosis induction via PI3K pathway inhibition. Consequently, this research underscores the potential of piperine as a natural anticancer agent for combatting human oral cancer.

The manuscript is well-structured and clearly written. The introduction, methodology, results, and discussion sections are logically organized, and the language is technically sound. The interpretation of the findings is thorough and aligns with the study's objectives. Based on the above assessment, I believe that the manuscript is well-prepared and meets the standards for publication in IJMS. I recommend its acceptance in present form.

Author Response

Response: We greatly appreciated the reviewer’s efforts to carefully review the paper. We also greatly appreciated the reviewer’s positive response to the paper.

Reviewer 2 Report

In this, authors showed that role of natural compound, piperine in oral cancer cell line, HSC-3 and its effects on tumor growth. HSC-3 cell line depicted reduced cell viability and increased apoptosis, autophagy upon treatment with piperine. It has been known that piperine acts as an anti-cancer agent in various tissue specific cancers including oral cancer. Addition of piperine to HSC-3 cell line inhibited the PI3K/Akt/mTOR pathway. Further it reduced oral cancer tumor growth in vivo mouse model of oral cancer suggesting that anti-tumor efficacy of piperine. Overall, this study was very well designed and performed to examine the role of piperine on autophagy and apoptosis in oral cancer cell line. However, the drawback of this study is using one oral cancer cell line. This was compensated by testing piperine in vivo xenograft model as well as the reported mechanism of action of this compound.

Minor editing of english language is needed.

Author Response

Response: We greatly appreciated the reviewer’s efforts to carefully review the paper. We also greatly appreciated the reviewer’s positive response to the paper. HSC-3 cells are a widely used cell line for oral squamous cell carcinoma research due to their highly aggressive properties. We confirmed the anticancer effects of piperine on HSC-3 cells in vitro and in vivo, supporting the potential of piperine against human oral squamous cell carcinoma.

Reviewer 3 Report

The manuscript ijms-2608217 is a exploratory study investigating the potential of piperine in head and neck cancer management. Briefly, the manuscript is well constructed, scientifically written, and it makes sense. The presentation is excellent. Figures are used appropriately and are of good quality, supported by raw data.

Few minor comments

1. L185 (and 421): Please indicate the number of mice per group used for the experiment. L421: Indicate the total number of mice required for this study.

2. L424: Are the sections cubic at 3 mm?

3. It should be mentioned that no primary tumor cell culture was tested. Could the authors repeat the results on non-immortalized cell lines (e.g. primary HNC)? If not, this should be added to the limitations of the study. 

4. For the perspectives (L337), please indicate whether more research is currently undertaken by the study group.

5. Please discuss piperine metabolism, e.g. considering per os administration.

6. Reference list: 35/62 references are older than 10 years. Please update whenever possible. First demonstration is acceptable (and encouraged in reviews), but an up-to-date reference list is recommended for the present manuscript. 

Reviewer 4 Report

Han et al aims to investigate the molecular mechanism of anticancer effect of piperine in HSC-3 human oral cancer cells. To this end, the authors examine the rate of apoptosis and autophagy. Then they check the status of the PI3K signaling pathway. By using PI3K inhibitors, they examine the dependency of apoptosis and autophagy on this particular pathway. Finally, the authors check the anticancer effect of piperine in a mouse xenograft model. They report that piperine activates apoptosis and autophagy based on the analysis of specific markers. They also report that the induction of apoptosis and autophagy depends on the PI3K signaling pathway. These cellular findings are confirmed in a mouse xenograft model.

Although the anticancer effect of piperine has been reported in numerous cancer models, the authors provide mechanistic data in an oral cancer cell line where there is a correlation between autophagy and apoptosis mediated by the PI3K signaling pathway. I believe that the data presented could be of interest to the researchers working in the field. I also should mention that the data is presented clearly and cohesively.

Major points:

1. Figure 2:

1a. Line 96: Please insert a statement as to how DAPI is used to demonstrate apoptosis. Any reference? Additionally, the magnification in Figure 1A is not high enough to see what is presented. Would it be possible to incude a figure of higher magnification?

1b. It would be nice to show the cleavage of caspase-3

1c. Please show statistics on Figure 2B.

Minor points:

1. Lines 193, 321, 339, 450 and 451; “in vivo”  and “in vitro”à italicize please

2. Figure 4B, on the graph, please fix “protetin” à protein
